# Small Bowel Endoscopic Features of Eosinophilic Gastroenteritis

**DOI:** 10.3390/diagnostics13010113

**Published:** 2022-12-30

**Authors:** Yu Sasaki, Yasuhiko Abe, Naoko Mizumoto, Eiki Nomura, Yoshiyuki Ueno

**Affiliations:** 1Department of Gastroenterology, Faculty of Medicine, Yamagata University, 2-2-2 Iida-Nishi, Yamagata 990-9585, Japan; 2Division of Endoscopy, Yamagata University Hospital, 2-2-2 Iida-Nishi, Yamagata 990-9585, Japan; 3Department of Gastroenterology, Sendai City Hospital, 1-1-1 Asutonagamachi, Taihaku-ku, Sendai 982-8502, Japan

**Keywords:** capsule endoscopy, balloon enteroscopy, eosinophilic enteritis, jejunum, ileum

## Abstract

Eosinophilic gastroenteritis (EoGE) is a rare digestive disorder characterized by eosinophilic infiltration of the stomach and intestines. In the diagnosis of EoE, it is extremely important to recognize distinctive endoscopic findings and accurately detect increased eosinophilia in gastrointestinal tissues. However, endoscopic findings of EoGE in the small intestine remain poorly understood. Therefore, we conducted a literature review of 16 eligible papers. Redness or erythema was the most common endoscopic finding in the small bowel, followed by villous atrophy, erosion, ulceration, and edema. In some cases, stenosis due to circumferential ulceration was observed, which led to retention of the capsule during small bowel capsule endoscopy. Although many aspects of small bowel endoscopic findings in EoGE remain elusive, the findings presented in this review are expected to contribute to the further development of EoGE practice.

## 1. Introduction

Eosinophilic gastrointestinal diseases (EGIDs) are uncommon entities characterized by abnormal accumulation of eosinophils in the gastrointestinal tract in the absence of secondary causes of eosinophilia, resulting in eosinophilic inflammation, gastrointestinal injury, and dysfunction [1,2,3]. Based on the extent of eosinophilic infiltration, EGIDs are classified as eosinophilic esophagitis (EoE), eosinophilic gastritis (EoG), eosinophilic gastroenteritis (EoGE), eosinophilic enteritis (EE), and eosinophilic colitis (EoC) [1,2,3]. Considering that EoGE, EoG, EE, and EoC are not clearly distinguished in practice, EoG, EE, and EoC are often included in the EoGE category [1,2,3].

EoGE, unlike EoE, is considered a rarer disease, however, EoGE is reported more frequently than EoE in Japan, with 5.5-fold higher incidence rate [4]. In the U.S., a large database study [5] between 2009 and 2011 showed that the standardized estimated prevalence of EoG, EoGE, and EoC were 6.3/100,000, 8.4/100,000, and 3.3/100,000, respectively. However, a more recent population-based study in the U.S., from 2012 to 2017, revealed a lower prevalence of EoGE (5.1/100,000) and EoC (2.1/100,000) [6].

Recently, EGIDs have become well recognized; neverthelessr, the actual incidence of EGIDs is difficult to estimate owing to underdiagnosis. The mucosa of the stomach and small bowel is more commonly involved, whereas colonic involvement is less frequent. Patients present with a variety of nonspecific symptoms that can delay diagnosis [3]. In a prospective study of 122 patients with refractory upper gastrointestinal symptoms for at least one month, seven patients (5.74%) had a missed diagnosis of EoGE [7]. The mean diagnostic delay for patients with EoGE in U.S. was reported to be 3.6 years, with delayed gastroenterologist referral, delayed endoscopy, and lack of biopsy and/or pathology contributing to this delay. Alternative diagnoses, including functional dyspepsia, irritable bowel syndrome, gastroesophageal reflux disease, or chronic gastritis, were common and associated with diagnostic delays [8]. A cross-sectional online survey of 202 board-certified gastroenterologists indicated that a limited number of gastroduodenal biopsy collections, particularly few from normal appearance mucosae, and failure to request tissue eosinophil counts might contribute to underdiagnosis of EoGE [9]. Therefore, for an accurate diagnosis of EGIDs, it is important to entertain a high degree of clinical suspicion for EGIDs, perform endoscopy, thoroughly evaluate the findings, and examine gastrointestinal mucosal eosinophilia on biopsy. However, while specific endoscopic findings for EoE are well known [10], the characteristics of the endoscopic findings of EoGE, especially those of small intestinal lesions, are largely unknown. Therefore, in this study, we reviewed the characteristics of the endoscopic findings of the small bowel in EoGE.

## 2. Materials and Methods

We conducted a PubMed search on 25 July 2022, using the search query: ((Endoscopy) AND ((small bowel) OR (small intestine))) AND (eosinophilic gastroenteritis). The search yielded 60 papers. In addition, our manual search of references cited in these papers identified nine papers. After assessing the eligibility through abstract view, 16 papers were selected for inclusion in this study (Figure 1). The eligible 16 papers comprised 13 single case reports [11,12,13,14,15,16,17,18,19,20,21,22,23], two case series [24,25], and one original paper [26].

Although this was not a systematic review, this short review of small bowel endoscopic findings in EoGE was conducted based on the flow diagram for the identification of studies of the PRISMA 2020 statement [27].

## 3. Enteroscopic Findings of EoGE

Thirteen case reports with available small bowel endoscopic findings yielded a male-to-female ratio of 7:6, ranging in age from 2 to 66 years with a median age of 37 years (Table 1). Reviewing these 13 cases together with the two case-series (Table 2), showed that the most common small bowel endoscopic finding was redness or erythema (45.4%). Redness or erythema was mostly patchy (Figure 2a) [13,14,17,18,21,25], and only one case was of the scattered type (Figure 2b) [11]. One report described the lesions’ color tone as salmon [17].

The second most frequently reported finding was villous atrophy (40.9%; Figure 2c) [14,16,18,21,22,25]. Erosion (Figure 2d) and ulceration (Figure 2e) were the third most frequently reported findings (27.2%). We previously noted that erosions in patients with EoGE are characterized by surrounding redness [24]. Linear ulcers on the terminal ileum [24] or circumferential ulcers [12,19,20,22] have been reported as small bowel ulcers in patients with EoGE. In three of the four reported cases of circumferential ulceration, it was located in the ileum [12,19,22], and stenosis was present in all four cases. Edematous mucosa of the small bowel (22.7%, Figure 2f) has also been documented in patients with EoGE [11,13,15,19]. The few reported cases include mucosal congestion [16], whitish exudate [18], short rounded edematous villi [19], and dark blue coloration of the deeper layers of the terminal ileum [23]. The case in which dark blue coloration was observed was a serosal-type EoGE, suggesting that this finding may have captured submucosal changes in the small bowel.

The distribution patterns of small intestinal lesions of EoGE in the aforementioned 13 cases [11,12,13,14,15,16,17,18,19,20,21,22,23] were as follows: extending throughout the small bowel in 4 cases, distributed only in the ileum in 4 cases, distribution only in the jejunum in 3 cases, jejunum to proximal ileum in one case, and proximal jejunum to ileum distribution in one case (Table 1). EoGE lesions in three cases included in one of the case-series [24] were reported to be distributed throughout the small bowel. Five of the six cases reported in another case-series [25] were described as ileal lesions. These findings suggest that EoGE lesions often involve the ileum, and a retrograde approach may be the first consideration when performing enteroscopy.

We found only one original study [26] that examined small bowel endoscopic findings of EoGE. This retrospective study included 123 patients with EoGE. The duodenal findings reported were friability (2%), erythema (2%), nodularity (3%), stricture (1%), and ulcer (6%). However, in all patients, findings in the jejunum and ileum were not endoscopically evaluated. This paper concludes that mucosal findings are normal in many cases of EoGE, and that tissue biopsy is important for diagnosis and determining treatment efficacy.

## 4. Biopsy Methods

In the 13 cases [11,12,13,14,15,16,17,18,19,20,21,22,23] and two case series [24,25] reviewed here, eosinophilic infiltration of the small intestinal mucosa was basically detected using target biopsy tissue from endoscopically detected lesions (Table 1). In one case [15], a random biopsy from a normal-looking mucosa demonstrated eosinophilic infiltration, while in another case [14], a similar biopsy failed to detect eosinophilic infiltration originating from an area where no findings were identified. It is important to accurately identify subtle findings such as erosions and actively perform biopsies to diagnose EoGE [24]. Endoscopic findings are not always associated with eosinophilic infiltration in the mucosa [25]. In a recent analysis of patients with moderate-to-severe gastrointestinal symptoms participating in a clinical trial of lirentelimab for EoG and/or eosinophilic duodenitis (EoD) [28], a high percentage of patients with EoG/EoD were detected by systematic endoscopic biopsy [29]. In this study, EoG diagnosis required presence of ≥30 eosinophils/high-power field (eos/hpf) in ≥5 hpfs and EoD required ≥30 eos/hpf in ≥3 hpfs. Eosinophils were patchy in gastric and duodenal biopsies and counting eosinophils in at least 8 gastric biopsies and 4 duodenal biopsies was required to identify a patient with EoG/EoD [29]. When systematic biopsy histology is performed for the diagnosis of patients with chronic gastrointestinal symptoms, it is important to differentiate EGIDs from functional gastrointestinal disorders such as functional dyspepsia and irritable bowel syndrome It is also critical to appropriately communicate with the pathologist and request for evaluation of eosinophilic infiltration.

## 5. Complication Related to Enteroscopic Evaluation for EoGE Diagnosis

All four patients with circumferential ulceration developed stenosis [12,19,20,22], three of which resulted in capsule endoscopy retention [12,19,22]. One of the three cases required partial resection of the ileum for capsule endoscopy retrieval [22]. In the other two cases, the capsule was fortunately eliminated after treatment with oral budesonide [12] and without clinical obstruction [20]. In capsule endoscopic evaluation of small bowel lesions in EoGE, careful attention should be paid to small bowel dilatation, and a patency capsule should be used to confirm gastrointestinal patency [30], even if there is the slightest suspicion of obstruction.

## 6. Conclusions

Endoscopic findings of EoGE include redness/erythema, villous atrophy, erosion, ulceration, and edema. Ulcers are often circumferential and stenotic, causing retention of the capsule during small bowel capsule endoscopy. Although this is the first review of small bowel endoscopic findings in EoGE, it is a summary of a very small number of case reports, and publication bias should be considered. These findings may not be specific to EoGE and, therefore, may have been missed in daily clinical practice. In the presence of a clinically suspicious background for EoGE, tissue biopsy should be performed, even if the findings are subtle or unclear.

The characteristics of small bowel endoscopic findings on EoGE remain unclear. It is hoped that this review will provide a valuable source for further clarification of the endoscopic findings of the small bowel and further elucidation of the pathogenesis of EoGE through more detailed studies with a larger number of cases.

## Figures and Tables

**Figure 1 diagnostics-13-00113-f001:**
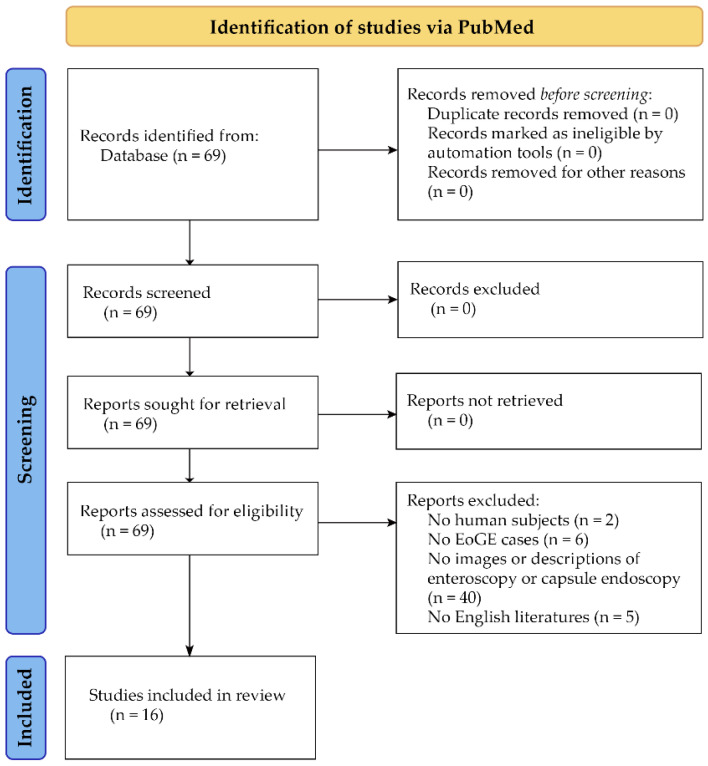
Flow diagram for identification of studies via PubMed.

**Figure 2 diagnostics-13-00113-f002:**
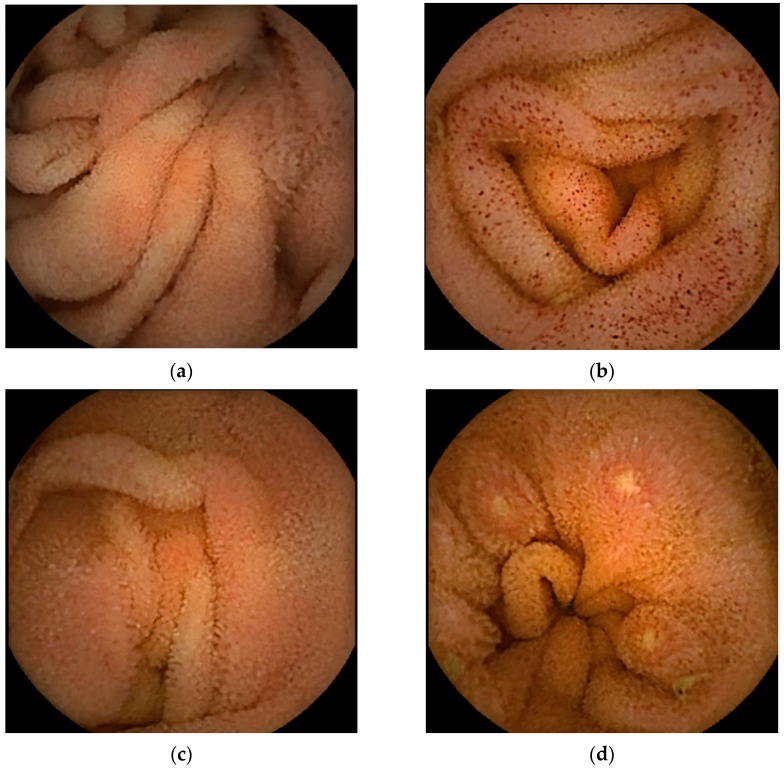
Representative images of small bowel capsule endoscopy of our own EoGE cases: (**a**) patchy redness with edema and atrophy; (**b**) scattered redness with edema; (**c**) villous atrophy with redness; (**d**) erosions with surrounding redness; (**e**) ulcers; (**f**) edema. All images were captured with PillCamSB3 (Covidien Japan Inc., Tokyo, Japan). EoGE: eosinophilic gastroenteritis.

**Table 1 diagnostics-13-00113-t001:** Enteroscopic findings of previously reported single case reports of EoGE patients.

Age/Sex Symptoms	WBC, μL (Eos%)	Enteroscopic Manifestations		Distribution of Small Bowel Lesions	Endoscopic Biopsy		Mucosal EI
		SCE	Enteroscopy		Site	Method	
20/M [11] Abdominal distention	20,480 (78.4)	ND	Mucosal hyperemia, edema, and scattered redness throughout the small intestines	Throughout	Duodenal bulb, small intestine, descending colon, and rectum	Target biopsy with enteroscopy	Present ^1^
28/F [12] Abdominal pain	Normal	Mucosal circumferential ulceration and partial stenosis	Same as SCE findings	Ileum	Ileal stenosis	Target biopsy with DBE	>52/HPF
45/F [16] Epigastralgia and dyspepsia	ND	Several focal areas of mucosal congestion, villous atrophy and pleomorphic denudation, and wide superficial erosions in the duodenum distal to the second portion	Focal villous atrophy and pleomorphic denudation	Proximal jejunum	Described just as small bowel, no details on specific site	Target biopsy with BE	>50/HPF
2/M [17] Fatigue and anorexia	ND (ND)	Jejunal and proximal ileal salmon-colored patches	ND	Jejunum to proximal ileum	Jejunum	Target biopsy with SBE	72/HPF
59/M [18] Nausea, abdominal pain, dysgeusia, weight loss, and diarrhea	ND (ND)	Patchy mucosal changes in the proximal jejunum	Erythema, scalloping, whitish exudate, and patches of villous blunting with a flat pink appearance were seen throughout the jejunum until the proximal ileum	Proximal jejunum	Duodenum, jejunum, and proximal ileum	Target and random biopsy with BE	>50/HPF
48/M [19] Abdominal pain and fullness	6900 (41)	Stenosis, dilation, and aperistalsis in the upper jejunum. Edematous mucosa, circumferential round ulcerated lesions and ulcer scars in the ileum	Edematous mucosa and stenosis in the ileum. Circumferential ulceration, short, rounded, and edematous villi	Proximal jejunum and ileum	Ileum	Target biopsy with DBE	Marked ^1^
32/M [20] Diarrhea and lower limb edema	ND (ND)	Ulcerated stenosis Capsule stagnation	Multiple ulcerated stenosis	Proximal jejunum	Jejunum	Target biopsy with SE	30/HPF
41/M [21] Intermittent, colicky abdominal pain and watery diarrhea	ND (49.4)	Multiple patchy erythematous mucosa lesions with a loss of villi throughout the small bowel	Same as SCE findings	Throughout	Described just as small bowel, no details on specific site	Target biopsy with enteroscopy	Present ^1^
32/M [22] Abdominal bloating and weight loss	ND (ND)	Ulcerated stenoses	Multiple ulcerated stenosis and partial villous atrophy	Ileum	Described just as small bowel, no details on specific site	Target biopsy with DBE	>50/HPF
37/F [23] Intermittent, colicky abdominal pain, vomiting, and abdominal distention	11,600 (31.0)	Dark blue coloration of the deeper layers of the terminal ileum	ND	Mid-ileum to the ileocecal valve	Terminal ileum	Target biopsy with colonoscopy	10–20/HPF
66/F [13] Abdominal pain and watery diarrhea	9370 (20.8)	ND	Slightly edematous and reddish mucosa in the jejunum and ileum	Throughout	Stomach, duodenum, jejunum, ileum, ascending colon, and rectum	Target biopsy with DBE	Present ^1^
47/F [14] Bilateral lower extremity edema	ND (20.7)	Multiple rounded patches of raised erythema with centrally flattened villi throughout the small intestine	Same as SCE findings	Throughout	Described just as small bowel, no details on specific site	Target biopsy with DBE ^2^	Present ^1^
24/F [15] Abdominal pain, watery diarrhea, and weight loss	16,600 (80)	ND	Mildly edematous mucosa in the terminal ileum	Ileum	Esophagus, stomach, duodenum, ileum, and colon	Random step biopsy with endoscopy ^3^	Present ^1^

^1^ The number of infiltrated eosinophils has not been described. ^2^ Push enteroscopy and colonoscopy with extensive random biopsy specimens from the esophagus, the stomach, the small intestine, and the colon were unremarkable. ^3^ Biopsies of the duodenum and ileum showed marked eosinophilic infiltration. Endoscopic findings of the esophagus, stomach, and colon were normal; however, eosinophilic infiltration was observed in biopsy specimens from all sites. WBC, white blood cell; Eos, eosinophil; EI, eosinophilic infiltration; M, male; F, female; SCE, small bowel capsule endoscopy; ND, not described; DBE, double balloon enteroscopy; BE, balloon enteroscopy; SBE, single balloon enteroscopy; SE, spiral enteroscopy; HPF, high-power field.

**Table 2 diagnostics-13-00113-t002:** Characteristics of enteroscopic findings of EoGE.

Enteroscopic Findings	Summary of 13 Single Case Reports	Case Series #1 [24] *n* = 3	Case Series #2 [25] *n* = 6	All *n* = 22
Redness/Erythema	7 (53.8)	0 (0)	3 (50.0)	10 (45.4)
Villous atrophy	5 (38.4)	0 (0)	4 (66.6)	9 (40.9)
Edema	4 (30.7)	0 (0)	1 (16.6)	5 (22.7)
Erosion	2 (15.3)	3 (100) ^2^	1 (16.6)	6 (27.2)
Ulceration	4 (30.7) ^1^	2 (66.6)	0 (0)	6 (27.2)
Stenosis	4 (30.7)	0 (0)	0 (0)	4 (18.1)
Capsule retention	3 (23.0)	0 (0)	0 (0)	3 (13.6)
Others ^3^	4 (30.7)	0 (0)	0 (0)	4 (18.1)

Values are presented as numbers (%). ^1^ All four cases were reported as circumferential ulcers. ^2^ All three cases were characterized by multiple small erosions with surrounding redness. ^3^ Other findings included mucosal congestion, whitish exudate, short-rounded edematous villi, and dark blue coloration. EoGE: eosinophilic gastroenteritis.

## Data Availability

Not applicable.

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
