# Peer review of "Small Bowel Endoscopic Features of Eosinophilic Gastroenteritis"

_diagnostics, 2022, doi:10.3390/diagnostics13010113_

Round 1
Reviewer 1 Report
This paper is an interesting review of endoscopic images of the small intestine in eosinophilic enteritis from previous literature.
Have you analyzed what was the distribution of small intestinal lesions in EGE? (e.g., where were the lesions more prevalent and In which areas was the disease stronger?)
Author Response
Response to Reviewer 1 Comments
Comment: This paper is an interesting review of endoscopic images of the small intestine in eosinophilic enteritis from previous literature.
Response: We appreciate your understanding of our work.
Point 1: Have you analyzed what was the distribution of small intestinal lesions in EGE? (e.g., where were the lesions more prevalent and In which areas was the disease stronger?)
Response 1: Thank you for your valuable suggestions. We have previously examined the distribution of small bowel lesions in EGE. The distribution of small bowel lesions in cases reviewed here have been added to Table 1 and are also included in the main text (page 3, lines 103–111).
Reviewer 2 Report
The article entitled ”SMALL BOWEL ENDOSCOPIC FEATURES OF EOSINOPHILIC GASTROENTERITIS” is an insightful investigation, a practical review, that brings new information regarding the structural alterations of a condition that is rather elusive. Eosinophilic gastrointestinal diseases are a group of heterogenous diseases characterized by eosinophilic infiltration of different parts of the gastrointestinal tract. While eosinophilic esophagitis is well-defined with established guidelines, clinically speaking, there is still much to learn about eosinophilic gastroenteritis or eosinophilic colitis. I find the conclusion of the article to be fair, giving the very small number of cases included in the analyses which, I consider to be the major drawback of the present work.
Major comments:
1. The introduction needs more effort. References regarding the last two sentences of the first paragraph need to be added. Also, the authors need to be more accurate on the epidemiology of the disease. Authors are kindly requested to emphasize the diagnostic difficulties and the reasons for the underdiagnosis of the disease. Please mention the biopsy methods that are used for diagnose.
2. Section 3 of the manuscript (“Enteroscopic findings of EGE”) needs substantial work on the structure of the sentences which are very short, and leave the impression of repetitiveness.
3. Table 1 needs extensive work. The sections “Treatment” and “Enteroscopic findings after treatment” should be removed taking into consideration that the authors do not address the treatment of the disease in the manuscript and some of your readers may draw wrong conclusions about the management.
4. Please mention and discuss the biopsy methods used in every study included in your analyses.
Minor comments:
1. Please rearrange the Figure 1 legend (Page 2, Line 58)
2. “Capsule retention” row of Table 2 on the section “Enteroscopic findings” should be removed
3. Last sentence of section 4 („Complication related to enteroscopic evaluation for EGE diagnosis”) needs reference.
Author Response
Response to Reviewer 2 Comments
Comment: The article entitled ”SMALL BOWEL ENDOSCOPIC FEATURES OF EOSINOPHILIC GASTROENTERITIS” is an insightful investigation, a practical review, that brings new information regarding the structural alterations of a condition that is rather elusive. Eosinophilic gastrointestinal diseases are a group of heterogenous diseases characterized by eosinophilic infiltration of different parts of the gastrointestinal tract. While eosinophilic esophagitis is well-defined with established guidelines, clinically speaking, there is still much to learn about eosinophilic gastroenteritis or eosinophilic colitis. I find the conclusion of the article to be fair, giving the very small number of cases included in the analyses which, I consider to be the major drawback of the present work.
Response: We appreciate your evaluation and understanding of our study. The following careful revisions have been made to the manuscript.
Major comments:
Point 1: The introduction needs more effort. References regarding the last two sentences of the first paragraph need to be added. Also, the authors need to be more accurate on the epidemiology of the disease. Authors are kindly requested to emphasize the diagnostic difficulties and the reasons for the underdiagnosis of the disease. Please mention the biopsy methods that are used for diagnose.
Response 1: Thank you for your comments. We have added citations to the last two sentences of the first paragraph. In the introduction, we have added the epidemiological data on EGIDs, their diagnostic difficulties, reasons for underdiagnosis, and biopsy methods (pages 1 to 2, lines 35–56). Regarding the biopsy methods, we have included the corresponding information for cases reviewed in the present studies in table 1 and in the main text.
Point 2: Section 3 of the manuscript (“Enteroscopic findings of EGE”) needs substantial work on the structure of the sentences which are very short, and leave the impression of repetitiveness.
Response 2: We have added a description of the distribution of small bowel lesions (page 3, lines 103–111). In addition, we have created and described a new section regarding the biopsy methods.
Point 3: Table 1 needs extensive work. The sections “Treatment” and “Enteroscopic findings after treatment” should be removed taking into consideration that the authors do not address the treatment of the disease in the manuscript and some of your readers may draw wrong conclusions about the management.
Response 3: Thank you for your suggestion. We have removed the “Treatment” and “Enteroscopic findings after treatment” columns from Table 1. In addition, we have omitted the corresponding description from the text (page 3, lines 100–102). Further, we have added two columns—“Distribution of small bowel lesions” and “Endoscopic biopsy”—in the Table 1.
Point 4: Please mention and discuss the biopsy methods used in every study included in your analyses.
Response 4: Thank you for your valuable suggestion. We have added the biopsy methods in Table 1 and the Section 4 (page 7, lines 141–161).
Minor comments:
Point 1: Please rearrange the Figure 1 legend (Page 2, Line 58)
Response 1: Thank you for your kind remark. We have rearranged Figure 1 legend so that is placed immediately following the Figure 1.
Point 2: “Capsule retention” row of Table 2 on the section “Enteroscopic findings” should be removed
Response 2: In line with this comment, we have removed the “Capsule retention” from Table 2.
Point 3: Last sentence of section 4 („Complication related to enteroscopic evaluation for EGE diagnosis”) needs reference.
Response 3: Thank you for your comment. The last sentence of Section 4 is our argument based on the results of this review, so there is no reference to the entire sentence; however, we have added a reference to the patency capsule.
Round 2
Reviewer 2 Report
The manuscript is suitable for publishing.